



# Development of a Spatial Hydrologic Soil Map Using Spectral Reflectance Band Recognition and a Multiple-Output Artificial Neural Network Model

Khamis Naba Sayl[1,3], Haitham Abdulmohsin Afan[1], Nur Shazwani Muhammad[1], Ahmed ElShafie[2]

[1]Department of Civil and Structural Engineering, Faculty of Engineering and Built Environment, Universiti Kebangsaan Malaysia, 43600 UKM Bangi, Selangor Darul Ehsan, Malaysia
[2]Department of Civil Engineering, Faculty of Engineering, University of Malaya, 50603 Kuala Lumpur, Malaysia
[3]Department of Dams and Water Resources, Engineering College, University of Anbar, Ramadi, Iraq

*Correspondence to*: Nur Shazwani Muhammad (shazwani.muhammad@ukm.edu.my)

**Abstract.** Soil type is important in any civil engineering project. Thorough and comprehensive information on soils in both the spatial and temporal domains can assist in sustainable hydrological, environmental and agricultural development. Conventional soil sampling and laboratory analysis are generally time-consuming, costly and limited in their ability to retrieve the temporal and spatial variability, especially in large areas. Remote sensing is able to provide meaningful data, including soil properties, on several spatial scales using spectral reflectance. In this study, a multiple-output artificial neural network model was integrated with geographic information system, remote sensing and survey data to determine the distributions of hydrologic soil groups in the Horan Valley in the Western Desert of Iraq. The model performance was evaluated using seven performance criteria along with the hydrologic soil groups developed by the United States Geological Survey (USGS). On the basis of the performance criteria, the model performed best for predicting the spatial distribution of clay soil, and the predicted soil types agreed well with the soil classifications of the USGS. Most of the samples were categorized as sandy loam, whereas one sample was categorized as loamy sand. The proposed method is reliable for predicting the hydrological soil groups in a study area.

## 1 Introduction

Spatial and temporal information on soil type is important in any civil engineering project in order to ensure sustainable hydrological, environmental and agricultural development. Hydrological processes, including surface runoff and infiltration, depend on the soil texture. Therefore, soil type is important in determining the potential volume of surface runoff and selecting the best type and location of water-harvesting structures (Jasrotia et al. 2009).

Currently, the Soil Conservation Service (SCS) method is used widely in many types of studies to estimate the surface runoff from a certain rainfall event (Senay and Verdin 2004; Winnaar et al. 2007; Tyagi et al. 2008; Elewa and Qaddah 2011). This method is based on the runoff curve number (CN), which is derived from the soil texture and land cover,



where soil type is classified into several subcategories. The United States Geological Survey (USGS) has divided
hydrologic soil groups into four classes. Runoff can be estimated using CN from the rainfall amount (Senay and
Verdin 2004). Many hydrological models use CN as input to estimate storm runoff, such as the Soil and Water
Assessment Tool (SWAT) (Neitsch et al. 2011), Environmental Policy Integrated Climate (EPIC) model (Wang et al.
2012) and Agricultural Non-Point Source Pollution (AGNPS) model (Young et al. 1987).

The identification of soil type based on laboratory testing is considered to be a traditional method, and it is time-
consuming and costly. Remote sensing (RS) represents one of the best alternatives; it is an accessible method that can
be utilized to provide valuable information related to site evaluation, including site monitoring and soil investigation.
Furthermore, this information can be easily analysed and integrated for site design, environmental impact assessment
and planning for construction activities. RS is capable of providing soil information in a spatial form, which is very
significant in the prediction of soil properties based on various bands of the electromagnetic spectrum.

The spectral reflectance characteristics of soils are a function of several important characteristics (Lacoste et al. 2014;
Martin et al. 2014; Wulf et al. 2014). The chemical and physical properties of materials define their spectral reflectance
and emittance spectra, which can be used to identify them. Spectral reflectance refers to the ratio of radiant energy
reflected to the incident energy on a body (Sims and Gamon 2002).

The soil reflectance data can be measured under laboratory conditions (i.e. proximal sensing) or in the field (i.e. RS).
The process of measuring soil reflectance using the proximal sensing method suffers from several problems, such as
variations in view angle, illumination, soil surface roughness and exact ground position. The effectiveness of RS
depends on the atmospheric conditions and the strength of the signal in the study area. The relationship between soil
type and reflectance is represented by five specific soil spectral reflectance curves developed by Stoner and
Baumgardner (1981). These curves provide important information about the presence or absence of organic matter
and iron along with absorption, which indicate different soil textures.

Over the last few decades, several studies have demonstrated that some soil characteristics can be determined using
laboratory spectral analysis (Salisbury and D'Aria 1992; Chang and Laird 2002; Nanni and Demattê 2006; Minasny
and McBratney 2008). Odeh and McBratney (2000) employed a multi-variate prediction model based on advanced
very-high-resolution radiometer (AVHRR) to map a large area of clay. The correlations between the image data and
laboratory analysis of SPOT, airborne spectroscopy and Landsat TM were used to determine different classes of soil
textures (Proctor et al. 2000). The various types of soil were classified with accuracy from 50% up to 100% using these
correlations. Such a poor correlation cannot be used to establish the relationship between soil texture and reflectance.
Chang and Islam (2000) used multi-temporal remotely sensed brightness temperature and soil moisture map to infer
the physical properties of soils. Two artificial neural networks (ANNs) were constructed based on the physical linkages
among the space–time distributions of brightness temperature, soil moisture and soil media properties.



Apan et al. (2002) used a primary component image of the Advanced Spaceborne Thermal Emission and Reflection
(ASTER) radiometer (bands 2 and 8) to determine classes of soil textures. They found that the absorption
characteristics of soil can be used to differentiate quartz and clay soils on the map. Chabrillat et al. (2002) found that
a short-wave with bands 5 and 6 of ASTER can detect clay soil, and the quartz index can be captured by thermal bands
10–14. Other studies showed that the short and thermal waves of ASTER can detect sandy and dark clayey soils,
although the results differed depending on the presence of organic substances (Salisbury and D'Aria 1992; Breunig
and Galvão 2008).

Soil reflectance is a complex phenomenon. It is difficult to predict the soil reflectance properties using physical models
and theories owing to the possibility of quantitative conversion of the reflectance spectrum of the multi-mineral surface
to the actual mineral abundances (Clark and Roush 1984). In addition, the theoretical results do not usually agree with
reality and are not valid for the assessment of soil properties (Dewitte et al. 2012; Wulf et al. 2014). Thus, we need to
establish a method that is able to reveal the complex relationships between reflectance and soil properties, especially
in large areas.

This study presents a methodology for the recognition of soil textures. This methodology integrates ANN, Geographic
Information System (GIS) and RS. Artificial intelligence (AI) improved the viewpoints of digital soil mapping, and
the integration of GIS helps to achieve complete area coverage. The proposed methodology is extremely useful in
large areas, where data are scarce and have limited availability. The applicability of this method is tested to a study
area located in the Western Desert of Iraq.
**2 Study Area**
Wadi Horan, which is one of the largest valleys in the Western Desert of Iraq, was selected as the study area. It is
located in the southern part of the Euphrates River, and its geographic coordinates range from 32° 10′ 44″ to 34° 11′
00″ N (latitude) and from 39° 20′ 00″ to 42° 30′ 00″ E (longitude), as shown in Fig. 1. The total catchment area is
13,370 km$^2$ (length = 362 km and width = 49.3 km). The perimeter and shape coefficients are 1,307 km and 0.13,
respectively.

The general climate is arid, which means that the area is dry in summer and cool in winter. There is a significant
variation in daily temperature (i.e. approximately 36°C). These conditions cause the land surface to heat up during the
day and cool during the night, which breaks the land surface into fragments and blocks. The high annual amount of
evaporation (3,200 mm), low average annual rainfall (115 mm) and high infiltration rate (3.25 mm/h) result in water
scarcity in the region.

The study area is flat, and the elevation increases moving westward. The average topographic incline from east to
west is 5 m/km, and the elevations of the highest and lowest points in the area are 987 and 77 m above sea level,



respectively. The main landscape is a plateau characterized by dense valleys. Some of the valleys are canyon-like with
lengths of a few tens of kilometres, and others are few hundred kilometres in length.

The major plateau of the catchment is rocky. The landform of the study area results from the complex interactions
among the structure, lithology and climate. The lithologic column of the uncovered rocks in the Western Desert
consists of limestone, dolomitic limestone, marl, dolomite, claystone, sandstone and phosphorite with rare gypsum
(Sissakian et al. 2011). In general, the Western Desert is characterized by low rainfall, thick soil cover and the absence
of vegetation. The study area includes some positive topographic features such as canyons, cliffs, depressions and
major valleys. Depressions, either erosional or solution in form, are another characteristic feature. The depressions
have different sizes and shapes, primarily including circular, oval and longitudinal. These features are the most
important components in building water-harvesting structures.
**3 Methodology**
The steps employed to accomplish the objectives of this study were data collection, data preparation and modelling,
as shown in Fig. 2. The details regarding the methodology are provided in the following paragraphs.

Satellite images of the study area were collected from Landsat 8 in August 2014. These images were imported into
the ERDAS Imagine software for geometric correction using WGS 84/UTM zone 38 projection. Subsequently,
unsupervised classification was carried out in the study area. Results from the unsupervised classification provide a
good depiction of some spectral classes and categorized these classes on the basis of the ranges of the image value.
Therefore, unsupervised classification is a useful task and includes the preparation of a primitive map for
reconnaissance, soil survey, to identify locations for soil sampling to reduce the effort time and cost. An easily
accessible flat surface consisting of bare soil and containing all types of soil with an area of $70 \times 70$ km$^2$ was selected
based on unsupervised classification of the entire study area. The primitive map was produced by colour-coding each
individual pixel to represent the class into which it was assigned by the classification algorithm. This map is a useful
way to present the information extracted by the classification process. In addition, the use of the primitive map will
reduce the error in pixel vegetation cover by more than 20% along with the errors associated with the spectral
signatures urban areas, water, roads, slope, soil roughness, locations and topography for each point selected. All these
specifications are recommended for accurate classification (Bartholomeus et al. 2008). Thus, the unsupervised
classification is an essential step in preparing the primitive maps, conducting the soil survey and collecting the soil
sample.

In the next phase, soil sampling locations were pre-selected based on the unsupervised classification thematic map.
Twenty-five sampling locations throughout the study area were selected using a GPS instrument based on certain
criteria. Subsequently, the soil samples were brought to the laboratory, and sieve analysis was carried out to estimate
the percentages of sand, clay and silt in each sample. The particle size analysis of a soil sample involves determining





percentage by weight of particles within different size ranges. The sieve analysis data were divided into training and
validation sets containing 19 and 6 samples, respectively.

Subsequently, site investigations were carried out. A satellite image from Landsat 8 was used to determine the spectral
reflectance of each location using ERDAS software based on the actual locations, which were determined using a GPS
device. The spectral reflectance for the visible, near infrared and short wave infrared, which are represented by nine
bands, were recorded for each location, whereas two thermal infrared bands were reduced.

After the laboratory work and site investigations were complete, a sensitivity analysis was carried out to examine the
relationship between bands and soil texture. The results were used to develop a database for soil type based on spectral
reflectance using the radial basis neural network model. The results of this model for each type of soil have been
evaluated based on seven criteria [i.e. root mean square error (RMSE), normalized root mean square error (NRMSE),
mean absolute error (MAE), normalized mean absolute error (NMAE), minimum absolute error, maximum absolute
error and correlation coefficient ($r$)]. The results of the radial basis neural network were verified using the hydrologic
soil group classification developed by USGS. The soil classifications were then manipulated within ArcGIS 10.2 using
the spatial analyst model to generate a digital map of hydrologic soil groups for the entire study area. Figure 3
summarises the methodology used in this study, including the strategies for data collection, data manipulation and
modelling.
**4 Results and Discussion**
The selection of sampling locations is based on certain criteria which have been mentioned previously in the
methodology to reduce the errors associated with spectral signatures and accurately estimate soil characteristics; thus,
a better unsupervised classification was performed, as shown in Fig. 4. Figure 4 shows ten classes of land cover
(vegetation and different types of soil), and each class is given a specific colour. The soil texture for each position is
given in detail in Table 1. The spectral reflectance was recorded for each position using ERDAS software. Nine bands
were used, as shown in Table 1.

A sensitivity analysis was carried out to validate the relationship between soil type and spectral reflectance, as shown
in Fig. 5. Soil type could not be detected by band 2 (wavelength (0.45–0.51) µm). Band 9 (1.36–1.38 µm) and band 7
(2.11–2.29 µm) were the most sensitive to soil type, particularly silt and sand, whereas clayey soil could be detected
by band 6 (1.57–1.65 µm), band 1 (0.43–0.45 µm) and band 7. Unfortunately, the spectral reflectance for each range
of wavelengths represented by the number of bands has a complex relationship with soil type because all these bands
participate in detecting the soil texture, but in different weights because of the mineral content of that soil. Because of
the variation in spectral reflectance over bands, a highly accurate model for the estimation of soil type is needed.
Therefore, it is important to include all effective bands in the ANN model.





The actual values and values estimated by the ANN model are given in Fig. 6, which shows that sand had a higher
percentage than silt and clay. Figure 6 also demonstrates that the values estimated for clay were more accurate than
those estimated for sand and silt, and the predicted value for sandy soil was significantly different from the actual
value. The overall performance of the ANN model was constant, and the total percentage of output (estimated) was

181    100%.


The performance of the ANN model for each type of soil was evaluated based on seven criteria, namely RMSE,
NRMSE, MAE, NMAE, minimum absolute error, maximum absolute error and *r*. The results indicate that the
estimation accuracy varied slightly among the three types of soil. The performance criteria for clay were excellent,
and the correlation coefficient for clay was the highest among the three soil types (Table 2). On the basis of the values
of NRMSE, NMAE and minimum absolute error, the ANN model generated less error for sand than for silt. In contrast,
based on RMSE, MAE, *r* and maximum absolute error, the silt estimation was better than the sand estimation.

Another way to evaluate the performance of the proposed method is through the hydrologic soil groups developed by
USGS, as shown in Fig. 7. The blue and red numbers in Fig. 7 are the measured and estimated soil textures,
respectively. The estimated values for all sites showed good agreement with the hydrologic soil groups of the USGS.
Most of the samples were categorized as sandy loam, whereas sample 1 was categorized as loamy sand. With reference
to Fig. 7, there was only a slight difference in the measured and estimated percentages of clay and sand for sample 1.
However, both the measured and estimated values are located in hydrologic soil group A. These results indicate that
the proposed method is reliable for predicting the soil group of a study area.

The hydrologic soil map was developed for Wadi Horan (Fig. 8). The spectral reflectance of 120 points in different
locations were predicted using the ANN model, and the percentages of soil types were determined based on the
hydrologic soil classifications of the USGS. Next, the classification data were manipulated within GIS using the spatial
analysis model to generate a digital map of hydrologic soil groups. Figure 8 shows the distributions of hydrologic soil
types in Wadi Horan Valley.
**5 Conclusions**
The integration of ANN with GIS, RS data and survey data helps to establish a significant procedure that can be
utilized for developing a digital hydrological soil group map. The relationship between spectral reflectance and soil
texture was used to predict soil class. This study proposed and evaluated a method to predict a hydrologic soil group
for an area by combining an ANN procedure with GIS and RS data. The effectiveness of this methodology was
evaluated based on seven performance criteria. The maximum absolute errors (one of the performance criteria) were
7.5, 12.8 and 14.8 for clay, silt and sand content, respectively. Clay soils produced the highest correlation coefficient
(0.8565). The overall performance of this methodology was also tested using the hydrologic soil groups developed by
USGS; all the samples were predicted to locate in the same hydrologic soil group determined by USGS. Therefore,




the proposed methodology performs well for classifying soils. It is also fast, reliable and cost-effective. In addition,
this method can be used to generate a database of high quality digital maps for authorities and stake holders.
**Author contribution**
K. N. Sayl collected the data, designed the methodology and perform the experiments. H. A. Afan and A. ElShafie
assist in the development of artificial neural network codes. N. S. Muhammad monitors the research progress and
prepared the manuscript with contributions from all co-authors.

**Acknowledgement**
The authors would like to thank Universiti Kebangsaan Malaysia for their financial support through the Geran Galakan
Penyelidik Muda, grant number GGPM-2014-046.

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

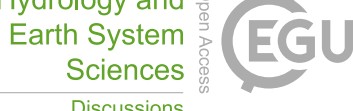


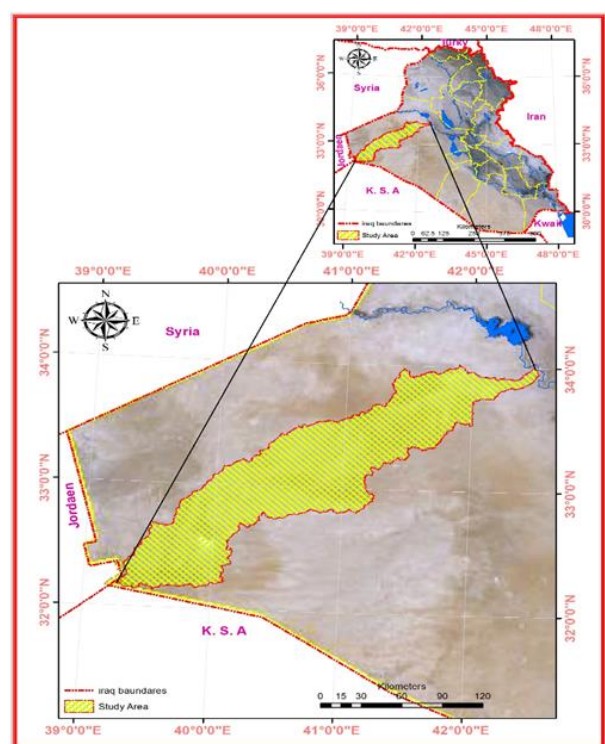

**Figure 1: Map of the study area**

**Figure 2: Schematic showing the study methodology**





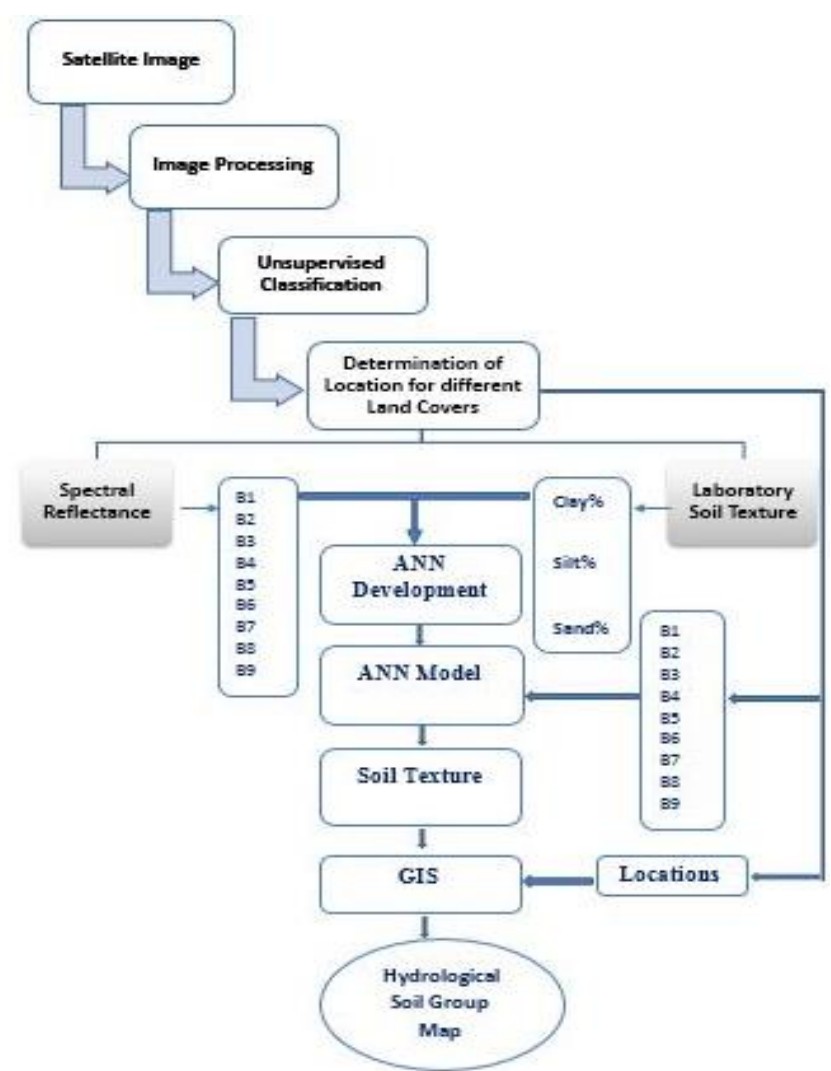

319                 **Figure 3: Flowchart showing the proposed methodology**




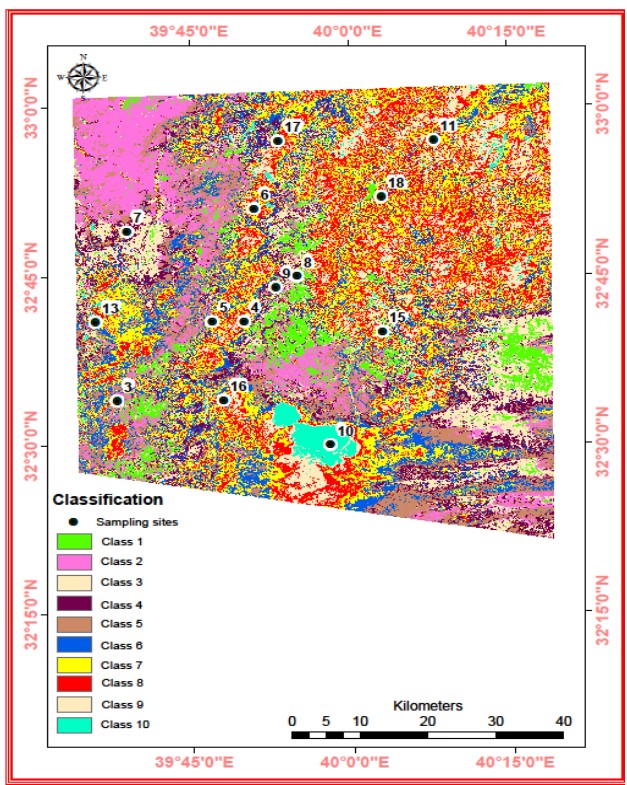


**Figure 4: Soil classes of unsupervised classification with samples locations**


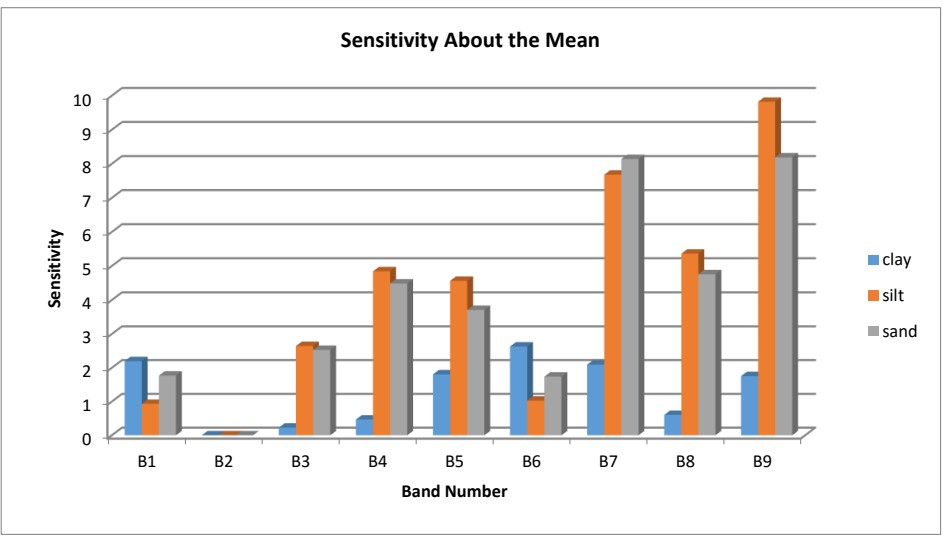


325                **Figure 5: Sensitivities of bands for different soil types**





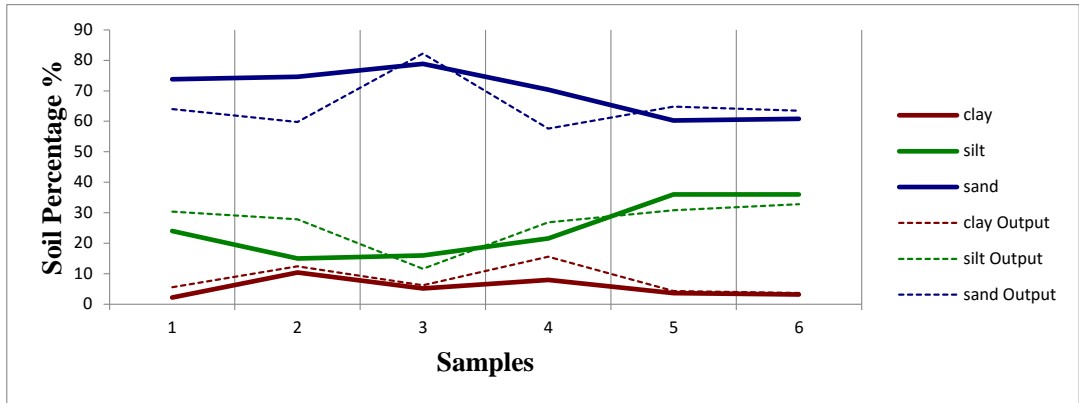

**Figure 6: Actual and estimated values of clay, silt, and sand for the tested samples**

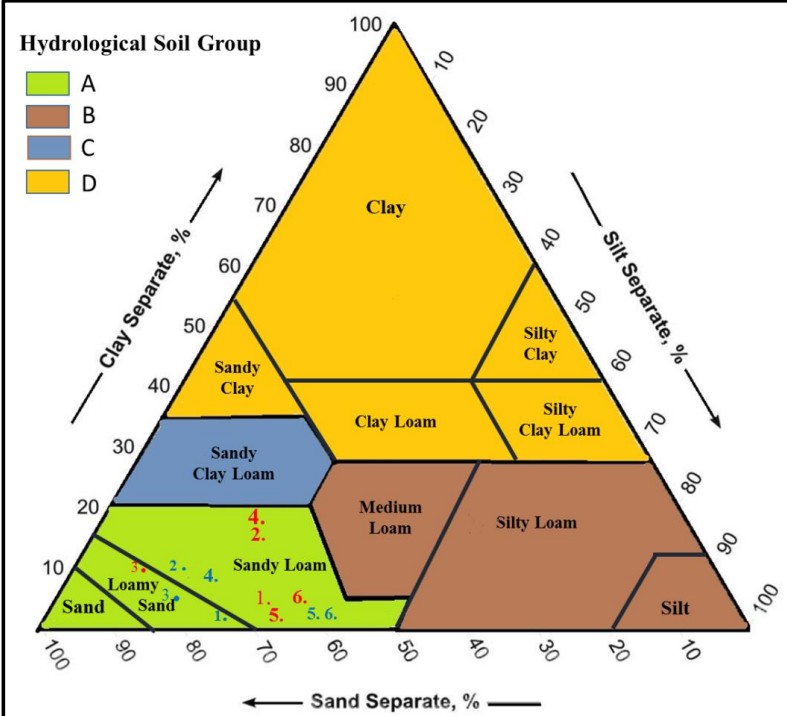

**Figure 7: Representations of actual and estimated points on the zones of the hydrologic soil group and the triangle of soil texture**





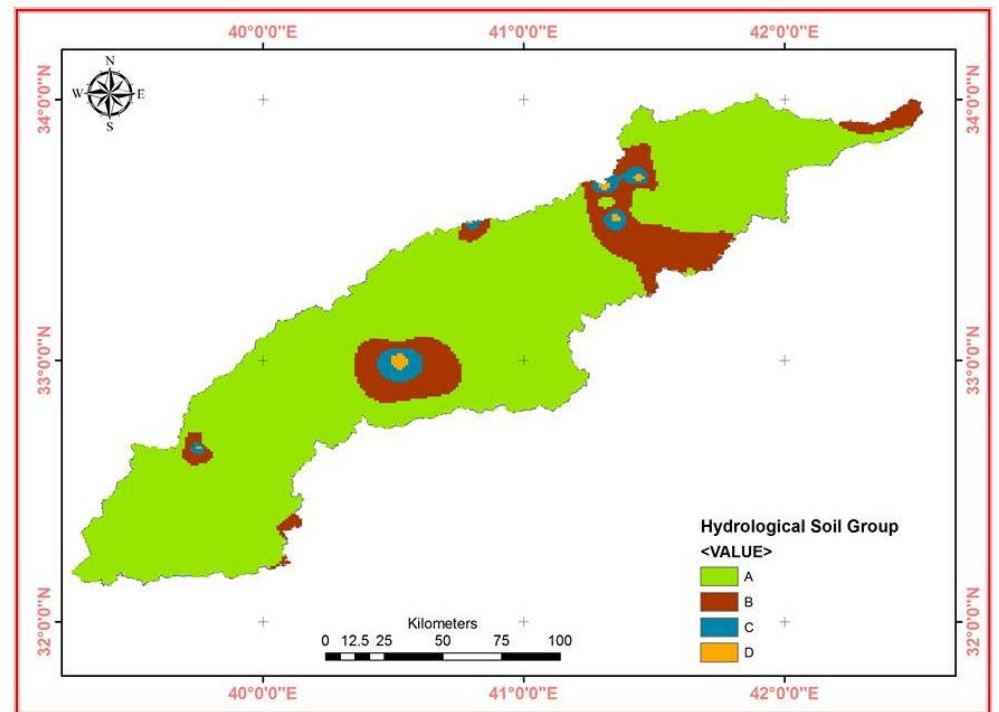


**Figure 8: Hydrologic soil group in the Wadi Horan Valley classified using the multiple-output artificial neural network model integrated with the geographic information system, remote sensing and survey data**

















351                    **Table 1: Band number and results of sieve analysis for each point**

| Point | Band 1 | Band 2 | Band 3 | Band 4 | Band 5 | Band 6 | Band 7 | Band 8 | Band 9 | Clay % | silt % | Sand % |
|---|---|---|---|---|---|---|---|---|---|---|---|---|
| P1 | 12664 | 12574 | 13864 | 18241 | 22641 | 26704 | 22598 | 15346 | 5106 | 31.2 | 21.2 | 47.6 |
| P2 | 12574 | 12498 | 13710 | 17721 | 21705 | 25666 | 21869 | 15103 | 5116 | 2.2 | 24.0 | 73.8 |
| P3 | 12936 | 12922 | 14485 | 18556 | 23386 | 27961 | 22736 | 15663 | 5094 | 10.4 | 15.0 | 74.6 |
| P4 | 13539 | 13839 | 16268 | 20959 | 25781 | 29141 | 23980 | 17721 | 5083 | 2.0 | 34.0 | 64.0 |
| P5 | 12897 | 12994 | 14743 | 19051 | 23350 | 26839 | 22378 | 16230 | 5080 | 2.6 | 23.0 | 74.4 |
| P6 | 13391 | 13626 | 15738 | 20259 | 24631 | 27881 | 23255 | 17193 | 5091 | 5.2 | 16.0 | 78.8 |
| P7 | 12909 | 12934 | 14478 | 18724 | 22853 | 25930 | 21894 | 15947 | 5079 | 1.0 | 7.6 | 91.4 |
| P8 | 12823 | 12866 | 14447 | 18179 | 22107 | 25529 | 20955 | 15423 | 5080 | 4.0 | 27.2 | 68.8 |
| P9 | 12802 | 12871 | 14446 | 18185 | 22275 | 25927 | 21041 | 15586 | 5108 | 8.0 | 21.6 | 70.4 |
| P10 | 15698 | 16683 | 20553 | 25715 | 30665 | 33089 | 27247 | 22258 | 5096 | 1.2 | 18.0 | 80.8 |
| P11 | 13208 | 13507 | 15929 | 20507 | 24550 | 27072 | 23370 | 17575 | 5093 | 7.2 | 8.0 | 84.8 |
| P12 | 12942 | 12924 | 14460 | 18642 | 22693 | 25803 | 21886 | 15917 | 5117 | 1.2 | 32.0 | 66.8 |
| P13 | 13163 | 13312 | 15123 | 19358 | 23521 | 26904 | 22438 | 16449 | 5112 | 3.7 | 36.0 | 60.3 |
| P14 | 13576 | 13799 | 16088 | 20704 | 25507 | 29300 | 24366 | 17540 | 5090 | 0.0 | 14.0 | 86.0 |
| P15 | 13127 | 13308 | 15479 | 19675 | 23761 | 26989 | 22313 | 16779 | 5085 | 4.0 | 39.2 | 56.8 |
| P16 | 13330 | 13551 | 15869 | 20805 | 25856 | 30040 | 24993 | 17481 | 5091 | 0.6 | 19.0 | 80.4 |
| P17 | 13227 | 13391 | 15504 | 19844 | 23964 | 26833 | 22087 | 16931 | 5086 | 3.2 | 36.0 | 60.8 |
| P18 | 13063 | 13290 | 15531 | 19598 | 23371 | 26544 | 22183 | 16370 | 5083 | 3.2 | 24.0 | 72.8 |
| P19 | 12664 | 12574 | 13864 | 18241 | 22641 | 26704 | 22598 | 15346 | 5106 | 50.2 | 1.5 | 48.3 |
| P20 | 12574 | 12498 | 13710 | 17721 | 21705 | 25666 | 21869 | 15103 | 5116 | 3.6 | 0.1 | 96.3 |
| P21 | 12936 | 12922 | 14485 | 18556 | 23386 | 27961 | 22736 | 15663 | 5094 | 32.0 | 1.6 | 66.4 |
| P22 | 13539 | 13839 | 16268 | 20959 | 25781 | 29141 | 23980 | 17721 | 5083 | 20.5 | 11.1 | 68.4 |
| P23 | 12897 | 12994 | 14743 | 19051 | 23350 | 26839 | 22378 | 16230 | 5080 | 33.5 | 0.8 | 65.7 |
| P24 | 13391 | 13626 | 15738 | 20259 | 24631 | 27881 | 23255 | 17193 | 5091 | 17.0 | 1.0 | 82.0 |
| P25 | 12909 | 12934 | 14478 | 18724 | 22853 | 25930 | 21894 | 15947 | 5079 | 31.2 | 1.1 | 67.7 |












**Table 2: Evaluation of the ANN model for each type of soil based on performance criteria**

| Performance criteria | clay | silt | sand |
|---|---|---|---|
| RMSE | 3.5221 | 6.9521 | 9.3021 |
| NRMSE | 0.1128 | 0.2201 | 0.212 |
| MAE | 2.5264 | 6.2112 | 7.9995 |
| NMAE | 0.0809 | 0.1965 | 0.1826 |
| Min Abs Error | 0.5295 | 3.2202 | 2.6906 |
| Max Abs Error | 7.5674 | 12.8539 | 14.8565 |
| $r$ | 0.8565 | 0.6471 | 0.4102 |
