# Peer review of "Development of a Spatial Hydrologic Soil Map Using Spectral Reflectance Band Recognition and a Multiple-Output Artificial Neural Network Model"

_Hydrology and Earth System Sciences, 2017_

## Referee Comment (RC1) · Anonymous Referee #1 · 8 Feb 2017

Sayl et al. 2017: Development of a spatial hydrologic soil map using spectral reflectance band recognition and a multiple-output artificial neural network.

The article deals with an important scientific development. The use of remote sensing for soil mapping. My main problem with the article is the extremely small dataset that the authors are using for their analysis. They used in total 25 soil samples, a small dataset that was split in 19 samples for validation and 6 for validation. Estimating an ANN on the basis of 19 samples that predicts sand, clay and silt contents on the basis of 9 bands of a LANDSAT 8 image just seems practically impossible or the authors should come with very good arguments. In addition, a validation on the basis of just 6 samples cannot provide accurate results. In addition, I have several other issues with

the paper: • The authors talk about "soils", "soil types", and "soil survey", but they only talk about topsoil texture in terms of sand, silt, clay. • line 54: proximal sensing can also be done under field conditions. • line 57: more important for efficient use of RS to characterize soil conditions is the soil cover (e.g., weeds, crop residues) and soil structure/roughness. • line 68: I agree that 50% is a poor correlation, but the 100% certainly is not a poor correlation. One should be more specific. • line 102-106: The authors indicated in the introduction that soil texture is an important property for the assessment of runoff. However, here the authors talk about a flat arid area, where the high infiltration rates are considered to be a problem. This does not seem to be very consistent. • line 110: if a large part of the plateau is rocky, this should be considered in the subsequent analysis. There is now reference how the authors dealt with the rocky area. • The authors should be more specific on the procedures. For example, they mention that the unsupervised classification was used to identify sampling locations but not how (line 138). They were certainly not selected by a GPS device. Most likely they were first identified on the classified map and subsequently located with the GPS. Another example, is that the soil classifications were "manipulated" with ArcGIS (line 156). However, no details on what the manipulation meant. • line 140. Distinguishing properly between silt and clay is practically impossible with sieving. Details on the sample treatment are required. • Line 152: Details on the Radial Basis Neural Network Model are required. • In general, you could say that the description of the methodology is insufficient to be able to even apply the methodology in another area. • Line 212: I wonder whether the authors can conclude on the basis of this study that the methodology is fast, reliable, and cost-effective.

Because of the above comments, I find the manuscript not suitable for publication in Hydrology and Earth System discussions.

---

## Author Comment (AC1) · 27 Feb 2017

The authors express sincere appreciation and thanked the Editorial Board and Reviewer #1 for the constructive comments. These comments will definitely improve the quality of the manuscript. The authors have addressed all of the concerns raised by Reviewer#1 in an itemized fashion below. The authors will include all suggestions in the revised version of the manuscript.

Reviewer #1: My main problem with the article is the extremely small dataset that the authors are using for their analysis. They used in total 25 soil samples, a small dataset that was split in 19 samples for validation and 6 for validation. Estimating an ANN on the basis of 19 samples that predicts sand, clay and silt contents on the basis of 9

bands of a LANDSAT 8 image just seems practically impossible or the authors should come with very good arguments. In addition, a validation on the basis of just 6 samples cannot provide accurate results.

Reply: The authors completely agreed with the reviewer that the used data for developing ANN model is somewhat small in terms of number of collected data, which may not be recommended for the development of an ANN model. The main objective of this research is to propose a methodology for the recognition of soil textures and proof it using available data. The authors believe that the changes in the soil texture at certain pilot area are expected to be relatively minor. However, more data would be more helpful at different sites of the study area. Therefore, the authors suggest to change the title of this manuscript to reflect the overall study, i.e. "Towards the Development of a Spatial Hydrologic Soil Map Using Spectral Reflectance Band Recognition and a Multiple-Output Artificial Neural Network Model".

Reviewer #1: The authors talk about "soils", "soil types", and "soil survey", but they only talk about topsoil texture in terms of sand, silt, clay.

Reply: The authors agreed with the reviewer on this issue. This study does focus on the topsoil texture in terms of sand, silt and clay because they represent the main variables of hydrological soil group. The authors will make sure that this term is consistent throughout the whole manuscript.

Reviewer #1: Line 54: proximal sensing can also be done under field conditions.

Reply: The following paragraphs will be added to give more details on this: "Most reported studies revealed the high potential of proximal sensing to estimate soil properties based on clear absorption features at the laboratory and local scale. However, for large scale mapping, this exercise need to be extended beyond the plot scale. Important qualitative and quantitative soil information can be obtained from remote sensing data."

Reviewer #1: Line 57: more important for efficient use of RS to characterize soil conditions is the soil cover (e.g., weeds, crop residues) and soil structure/roughness.

Reply: Unsupervised classification was adopted in the first step of the methodology to determine vegetation cover. The Normalized Difference Vegetation Index (NDVI) is used primarily for vegetation identification and to determine the lushness of vegetated land surfaces. Also, as mentioned in lines 122-126, "The methodology has been used the satellite image of the study area, represent arid region, (Landsat 8, August 2014) has been taken in dry season when the barren soil covers large area.

Reviewer #1: Line 68: I agree that 50% is a poor correlation, but the 100% certainly is not a poor correlation. One should be more specific.

Reply: The authors thank the reviewer for this valuable comment. This line is typo error. The authors have decided to delete this line to avoid any confusion re garding the correlation index.

Reviewer #1: Line 102-106: The authors indicated in the introduction that soil texture is an important property for the assessment of runoff. However, here the authors talk about a flat arid area, where the high infiltration rates are considered to be a problem. This does not seem to be very consistent.

Reply: The west desert of Iraq is classified as an arid region, with an uneven distribution of precipitation in time and space. The authors believe that the quantity of effective runoff is much more important as compared to the amount of annual rainfall. Most of the precipitation events in the study area are short duration and high intensity that occurred for short periods in a year. As a result, there is a call for more efficient water conservation alternatives.

Reviewer #1: Line 110: if a large part of the plateau is rocky, this should be considered in the subsequent analysis. There is now reference how the authors dealt with the rocky area.

Reply: The authors thank the reviewer for his comment. Actually, the rocky area is very small as compared to the overall study area. The authors will improve this statement in order to indicate that the rocky part is ignored in the analysis due to the small percentage and hence it has insignificant influence on the results.

Reviewer #1: The authors should be more specific on the procedures. For example, they mention that the unsupervised classification was used to identify sampling locations but not how (line 138). They were certainly not selected by a GPS device. Most likely they were first identified on the classified map and subsequently located with the GPS. Another example, is that the soil classifications were "manipulated" with ArcGIS (line 156). However, no details on what the manipulation meant.

Reply: As mentioned in lines 124-127, the main purpose of unsupervised classification is to select the soil sampling locations on a primitive map based on a good depiction of some spectral classes and they are classified by colour. Then, we used GPS device to identify these locations on site. The following paragraphs will be added to give more information about manipulation: " Arc GIS spatial analyst extension can convert the themes, depending on vector features to grids. Additionally, grids can be derived from various spatial analysis operations, and it is added to be viewed as grid themes. These grid cells have been classified in various ways and different colors are chosen for each class where the colors represent the progression of values for a data attribute specified. It is achieved after the raster themes are converted into a shape file, which includes the environmental characteristics that represents the hydrological soil group"

Reviewer #1: Line 140. Distinguishing properly between silt and clay is practically impossible with sieving. Details on the sample treatment are required.

Reply: The authors agreed with the reviewer that this part is unclear. The authors have included the following paragraph in order to give more information on the difference between silt and clay in this study. "The distribution of particle sizes larger than $75\mu$m (retained on the No.200 sieve) is determined by sieving, while the distribution of particle
sizes smaller than 75 $\mu$m is determined by sedimentation process, using a hydrometer to get the necessary data".

Reviewer #1: Line 152: Details on the Radial Basis Neural Network Model are required.

Reply: Authors agree with the reviewer that additional information on radial basis neural network is required. The following details will be added to the revised manuscript: "Radial Basis Neural Network (RBNN) is an artificial method that based on the interpolation of a multivariate function. RBNN consists of three layers, i.e. input layer for feeding feature vector to the network, hidden layer where the calculation of outcome of basis function is processed, and finally the output layer for linear combining the basic functions. The following figure shows the structure of RBNN.

Figure (). Structure of Radial Basis Function Neural Network

The hidden layer applies a non-linear transformation from the input space to the hidden space. The output layer applies a linear transformation from the hidden space to the output space. The radial basis functions $\varphi$ 1, $\varphi$ 2, .... $\varphi$ N are known as hidden functions while ãĂŰ{$\varphi$i(x)}ãĂŮ_(i=1)ˆN is called the hidden space. The number of basic functions (N) is typically less than the number of data points available for the input data set. Among several radial basis functions, the most commonly used is the Gaussian, which in its one-dimensional representation takes the following form: $\varphi$(x,$\mu$)=eˆ(-ãĂŰ∥x-$\mu$∥ãĂŮˆ2/ãĂŰ2dãĂŮˆ2 ) where $\mu$ is the center of the Gaussian function (mean value of x) and d is the distance (radius) from the center of $\varphi$(x,$\mu$), which gives a measure of the spread of the Gaussian curve.

The hidden units use the radial basis function. If a Gaussian function is used, the output of each hidden unit depends on the distance of the input x from the center $\mu$. During the training procedure, the center $\mu$ and the spread d are the parameters to be determined. It can be deduced from the Gaussian radial function that a hidden unit is more sensitive to data points near the center. This sensitivity can be adjusted by controlling the spread d. It can be observed that the larger the spread, the less sensitivite radial basis function to the input data. The number of radial basis functions inside the hidden layer depends on the complexity of the mapping to be modeled and not on the size of the data set, which is the case when utilizing multi-layer perceptron ANN. Moreover, RBNN has the ability to recognize a complex relation between the input and output of the model. This research identify the relationship between the bands and soil types. RBNN model requires some important parameters to be established before perform the training process, such as the performance goal of 0.0005 and the spread constant of 1."

Reviewer #1: In general, you could say that the description of the methodology is insufficient to be able to even apply the methodology in another area.

Reply: Owing to the reviewer feedback, the authors have significantly improved the methodology section in order to include more detailed informat ion on the proposed methodology for developing a hydrological soil map. The methodology section in the revised manuscript has been split into two sub-sections, first, the Unsupervised Classification Process and the second part for the Radial Basis Function Neural Network (RBNN) method as pattern recognition technique. As reported in the previous comment, the authors added more detailed information on RBNN model. In addition, the authors will the following paragraphs in order to provide more details on the unsupervised classification process performed in this study: "Classification is the process of sorting pixels into a finite number of classes or categories of data based on their data file values. If a pixel satisfies a certain set of criteria, then it is assigned to the class that corresponds to that set of criteria. A pixel may be charÂňacterized by its spectral signature, which is determined by the relative reflectance in the different wavelength bands. Multispectral classification is an information-exÂňtraction process that analyzes these spectral signatures and then assigns pixels to categories based on similar signatures. There are two methods to classify pixels, i.e. unsupervised and supervised classification. Unsupervised classification identifies clusters or groupings in a feature space. A cluster is a set of points in the feature space where their local density is

large (relative maximum) compared to the density of feature points in the surrounding region. Techniques are useful for image segmentation and for classification of raw data to establish classes and prototypes. Clustering is also a useful vector quantization technique for compression of images. The clustering algorithm has no regard for contiguity of pixels that define each cluster. One of the most known unsupervised classifier method is the ISODATA. ISODATA is an iterative process where it repeatedly performs an entire classification (outputting a thematic raster layer) and recalculates the statistics. Self-Organizing refers to the way in which it locates clusters with minimum user input. Each iteration calculates means and reclassifies pixels with respect to the new means. Iterative class splitting, merging, and deleting are done based on the input threshold parameters. All pixels are classified to the nearest class unless standard deviation or distance threshold is specified. One of the primary advantages of unsupervised classification is many of the classes are created automatically." The authors believed that with this in-depth information in the methodology, including the added details on the unsupervised classification process and the radial basis function neural network (RBNN) , this section has improved significantly and could be applied in similar study area.

Reviewer #1: Line 212: I wonder whether the authors can conclude on the basis of this study that the methodology is fast, reliable, and cost-effective.

Reply: The authors conclude such benefits of the proposed methodology because the study area is located in a remote area and developing country such as Iraq. As mentioned in line 96, the total catchment area is 13370 km2. For such a large and remote area, there are limited infrastructures and financial resources. Additionally, there is a need to collect soil samples from this area which in other word means that it is time-consuming, unreliable and expensive. Therefore, the authors consider that the proposed methodology with its level of accuracy could be faster, reliable and cost-effective as compared with the time required to collect the soil samples for this remote and relatively large area. In addition, the proposed method could be more

reliable due to the untruthfulness of the collected soil sampling in such remote area and disfigurement for the samples due to transferring them from remote area to the laboratory for analysis. Finally, the authors consider that the proposed methodology for identifying the hydrological soil map is cost-effectiveness as compared with the standard procedure because collecting samples from such remote areas is very expensive, in addition to the cost of analyzing the samples. The authors improved the conclusion to include more details as reported here in order to be more accurate on the benefits of the proposed methodology and its advantages over the standard procedure at a similar study area.

Please also note the supplement to this comment:
http://www.hydrol-earth-syst-sci-discuss.net/hess-2017-13/hess-2017-13-AC1-supplement.pdf
* * *
[Figure]

**Fig. 1.** Structure of Radial Basis Function Neural Network

---

## Referee Comment (RC2) · Anonymous Referee #2 · 20 Mar 2017

The general approach of utilizing remote sensing for soil texture mapping in desert regions is quite valuable and has been demonstrated in the literature. I think the authors were right to look at ways to use these methods to try and more efficiently maps soils in Iraq. However, I have quite a few concerns about the manuscript submitted by Sayl and coauthors. I'm attaching the original pdf with specific comments which provide all specifics related to the general comments below.

1) Clarity of methods section As written, the methods section is not repeatable. Much effort is needed to be more specific in describing steps taken and cite all the algorithms used. Couple examples: 1) soil sampling technique (e.g. auger, open pit) and sampling depths were never specified, 2) radial basis neural network has no citations, and 3) the

[Figure]

unsupervised classification method is not specified or cited, and the software used is also not specified. There are quite a few other major methodological details where I got lost and couldn't figure out what was done (please see the commented pdf for full details).

2) Sampling design, size and inference The sample size (n=25; 15 training, 6 validation) was quite small for such a large area. The chosen smaller area for sampling also did not appear to represent the greater study area (flat accessible area versus a plateau with a dense network of valleys and canyons). The validation set represented a very small range of both soil texture separates (sand, silt, clay) and only fell within one of the USGS hydrologic groups. This limits the inference space to just that group and makes any claims about predicting the other groups correctly unsupported by the data and result. This makes extrapolation from the smaller sample area to the greater study area unsubstantiated.

3) Overall grasp of literature The literature review and breadth of topics covered and utilized suggest that the authors should consider expanding the tools and data they use for making these predictions (beyond having a better sample). For example, in the broad body of digital soil mapping studies, topographical layers from DEM, climate surfaces, and other spectral data (e.g. gamma radiometrics) are often the most effective predictors. I think including DEM variables could prove very effective at this scale, yet this was not done. I'm also not sure why this form of neural networks (NNs) was utilized. Generally random forests have been outperforming NNs, why not try other algorithms. Also, if hydro group is the desired target variable, why not predict that parameter? Machine learning seems to do better at classification (particularly random forests).

Based on these issues, I do not think this manuscript is appropriate for publication in HESS.

Please also note the supplement to this comment:

http://www.hydrol-earth-syst-sci-discuss.net/hess-2017-13/hess-2017-13-RC2-supplement.pdf

[Figure]

**Supplement:**

[revised manuscript text omitted]

---

## Author Comment (AC2) · 10 Apr 2017

The authors express sincere appreciation and thanked the Editorial Board and Reviewer #2 for their constructive comments and suggestions. These comments will definitely improve the quality of the manuscript. The authors have addressed all the concerns raised by Reviewer#2 in an itemized fashion below. The authors will include all suggestions in the revised version of the manuscript.

Reviewer #2: General Comments 1) Clarity of methods section, as written, the methods section is not repeatable. Much effort is needed to more specific in describing steps taken and cite all algorithms used. Couple examples: 1- soil sampling technique (e.g. auger, open pit) and sampling depths were never specified, 2- radial basis neural

network has no citations, 3- the unsupervised classification method is not specified or cited, and the software used not specified. There are quite a few major methodological details where I got lost and could not figure out what was done (please see the commented pdf for full details. Reply: The authors agree with the reviewer that more description on the methodology should be included in the manuscript. Therefore, the following paragraph that gives more details on soil sampling will be added to the revised manuscript: "There are twenty five (25) sampling locations throughout the study area. The selection of these sampling locations was made based on its accessibly, vegetation cover, the spectral signatures urban areas, water, slope, roads, soil roughness, location and topography. On-site, a GPS instrument was used to locate the sampling points. The soil samples were collected using auger and the depths were between 20 to 40 cm of the topsoil. Subsequently, the soil samples were brought to the laboratory for particle size distribution analysis. This analysis was done to determine the soil texture, which is an important parameter for hydrological behavior of soil. Particles above 50 $\mu$m were separated using wet sieving techniques, while small particles were analyzed using hydrometer method (Ryan et al. 1996; Skopp 2000). Soil particles are dispersed in water and the reduction of fluid density due to setting is determined through hygrometer (Skopp 2000). The particle size distributions of all collected soil samples were classified according to the USGS method texture classification.

The following paragraphs will be added in order to provide more details on the neural network and unsupervised classification process performed in this study: "Image classification method was used in order to extract the most relevant information needed to achieve the objectives of this study. This technique includes the process of sorting pixels into a finite number of classes or categories of data based on their data file values (Jain 1989) If a pixel satisfies a certain set of criteria, then it is assigned to the class that corresponds to that set of criteria. A pixel may be charǍňacterized by its spectral signature, which is determined by the relative reflectance in the different wavelength bands. Multispectral classification is an information-exǍňtraction process that analyzes these spectral signatures and then assigns pixels to categories based on similar signatures. One form of classification is unsupervised classification (Acharya et al. 2005). Unsupervised classification was carried out using the ERDAS Imagine software and it involves the identification of clusters or groupings in a feature space. A cluster is a set of points in the feature space where their local density is large (relative maximum) compared to the density of feature points in the surrounding region. Unsupervised classification permits an unbiased assessment of the total of the raw data. It can be used on the first hand to identify the main classes and then check the information in the field (Richards and Jia 1999). This method is preferred if area is very large and field data is lacking. It illustrates that a priori knowledge of the human observer is needed to assign the pixels to classes. One of the most known unsupervised classifier method is the Iterative Self Organizing Data Analysis (ISODATA). ISODATA is an iterative process where it repeatedly performs an entire classification (outputting a thematic raster layer) and recalculates the statistics. Self-Organizing refers to the way in which it locates clusters with minimum user input (Acharya et al. 2005). Each iteration calculates means and reclassifies pixels with respect to the new means. Iterative class splitting, merging, and deleting are done based on the input threshold parameters. All pixels are classified to the nearest class unless standard deviation or distance threshold is specified. One of the primary advantages of unsupervised classification is many of the classes are created automatically (Jain 1989)" The following information on Radial Basis Neural Network (RBNN) will be added to the revised manuscript: "Radial Basis Neural Network (RBNN) is an artificial method that based on the interpolation of a multivariate function (Lowe and Broomhead 1988) . RBNN consists of three layers, i.e. input layer for feeding feature vector to the network, hidden layer where the calculation of outcome of basis function is processed, and finally the output layer for linear combining the basic functions. The following figure shows the structure of RBNN.

Figure 1 Structure of Radial Basis Function Neural Network

The hidden layer applies a non-linear transformation from the input space to the hidden space. The output layer applies a linear transformation from the hidden space to the

output space. The radial basis functions $\varphi 1$, $\varphi 2$, .... $\varphi N$ are known as hidden functions while $\tilde{a}\breve{A}\acute{U}\{\varphi i(x)\}\tilde{a}\breve{A}\mathring{U}\_(i=1)^N$ is called the hidden space. The number of basic functions (N) is typically less than the number of data points available for the input data set. Among several radial basis functions, the most commonly used is the Gaussian, which in its one-dimensional representation takes the following form: $\varphi(x,\mu)=e^{(-\tilde{a}\breve{A}\acute{U}\hat{a}\acute{L}\breve{e}x-\mu\hat{a}\acute{L}\breve{e}\tilde{a}\breve{A}\mathring{U}^2/\tilde{a}\breve{A}\acute{U}2d\tilde{a}\breve{A}\mathring{U}^2}$ ) where $\mu$ is the center of the Gaussian function (mean value of x) and d is the distance (radius) from the center of $\varphi(x,\mu)$, which gives a measure of the spread of the Gaussian curve. The hidden units use the radial basis function. If a Gaussian function is used, the output of each hidden unit depends on the distance of the input x from the center $\mu$. During the training procedure, the center $\mu$ and the spread d are the parameters to be determined (Moody and Darken 1989). It can be deduced from the Gaussian radial function that a hidden unit is more sensitive to data points near the center. This sensitivity can be adjusted by controlling the spread d. It can be observed that the larger the spread, the less sensitive radial basis function to the input data. The number of radial basis functions inside the hidden layer depends on the complexity of the mapping to be modeled and not on the size of the data set, which is the case when utilizing multi-layer perceptron ANN. Moreover, RBNN has the ability to recognize a complex relation between the input and output of the model. This research identifies the relationship between the bands and soil types. RBNN model requires some important parameters to be established before perform the training process, such as the performance goal of 0.0005 and the spread constant of 1."

2a) Sampling design, size and the inference the sample size (25; 19 training, 6 validation) was quite small for such a large area. Reply: The authors completely agree with the reviewer that the data used to develop the ANN model is somewhat small in terms of the number of collected data, which may not be recommended for the development of an ANN model. However, the main objective of this research is to propose a methodology for the recognition of soil textures and validate it using data collected on-site. The authors believe that the changes in the soil texture at certain locations within the study area are expected to be relatively minor. More data would be more

helpful at different sites of the study area. Therefore, the authors suggest to change the title of this manuscript to reflect the scope of this study, i.e. "Towards the Development of a Spatial Hydrologic Soil Map Using Spectral Reflectance Band Recognition and a Multiple-Output Artificial Neural Network Model".

2b) The chosen smaller area for sampling also did not appear to represent the greater study area (flat accessible area versus a plateau with a dense network of valleys and canyons).

Reply: The study area (Wadi Horan) is a part of West desert of Iraq, therefore the authors believe that the changes in soil texture at certain areas are expected to be relatively minor. As mentioned in the manuscript, Wadi Horan is flat, and the average topographic incline from east to west is 5 m/km. The plateau with a dense network of valleys and canyons is very small as compared to the overall study area. Therefore, the authors decided to clarify this characteristic of Wadi Horan in the revised manuscript to avoid any confusion.

2c) The validation set represented a very small range of both soil texture separates (sand, silt, clay) and only fell within one of USGS hydrologic group. This limits the inference space to just that group and makes any claims about predicting the other groups correctly unsupported by the data and result. This makes extrapolation from the smaller sample area to the greater study area unsubstantiated. Reply: The authors completely agree with the reviewer that this study considered all soil types in the USGS hydrological group during the training stage of model, i.e. as input data shown in Table 1. However, the validation process did not indicate all types of soil in the USGS hydrological soil group even though the regression model has the ability to extrapolate it during the training stage. In this study, the validation exercise resulted in certain types of soil but it does not mean that model is unable to predict other soil types. The model simulation shows that all hydrological soil group types were recognized in the study area, as shown in figure 8.

3a) Overall grasp of literature review and breadth of topics covered and utilized suggest that authors should consider expanding the tools and data they use for making these predictions (beyond having a better sample). For example, in the broad body of digital soil mapping studies, topographical layers from DEM, climate surfaces, and other spectral data (e.g. gamma radiometrics) are often the most effective predictors. I think including DEM variables could prove very effective at this scale, yet this was not done.

Reply: The authors agreed with the reviewer that including DEM variables could be very effective at this scale. Also, climate surface, and other spectral data are the most effective predictors. The authors considered the topographical layers from DEM, the Normalized Difference Vegetation Index (NDVI) in dry and wet season, and climate data as shown in line 128. These data extracting techniques are well known and commonly used in research (Cvar 2014; Sayl et al. 2016). The following paragraphs will be added to give more information on the tools and data used in making these predictions (beyond having a better sample): "The DEM generated from Shutter Radar Topographic Mission SRTM data was used to present the topography of the study area (as shown in Figure 2 below), slope data and drainage. The earth features, such as the Normalized Difference Vegetation Index (NDVI) is used primarily for vegetation identification and to determine the lushness of vegetated land surfaces (using ERDAS Imagine software). These features were considered in this study as a basic guide during the field work. In addition, the location of the points used to generate digital soil map were selected according to DEM generated for a small area in order to overcome the problem of scale".

Figure 2 Topography of the location used to generate digital soil map and DEM of the study area generated from Shutter Radar Topographic Mission data

3b) I am also not sure why this form of Neural Network (NNs) was utilized. Generally random forests have been outperforming NNs, why not try other algorithms. If hydro group is the desired target variable, why not predict that parameter? Machine learning seems to do better at classification (particularly random forests).

Reply: The authors would like to highlight that this study is the first attempt to develop an Artifical Intelligence (AI) model for hydrological soil group. There are several AI methods characterized by the efficiency in prediction that can be utilized for random forests and this may be considered in future research. However, the Radial basis neural network (RBNN) was chosen in this study because this method simple and readily available in the Matlab tool box. Additionally, RBNN proved its ability to provide multiple output with a good accuracy in this study. In this study, the regression model could be further generalized as compared to the classification model. This model built a relationship between the reflectance bands with soil percentages and it is very sensitive. During classification process, the model in training must include all classes and due to the limitation of data set in this study, it is recommended to use regression model than classification. Therefore, this model will be more efficient to find other types of soil based on the output percentages of soil type even if it is not included in training set. Hence, this model setup is intended to be more flexible.

Specific comments

1) Line 20-24 p.1- Hard to tell what was actually mapped? Texture class? Clay%? Reply The main objective of this study is to determine the distribution of hydrological soil groups in the study area. This is based on the percentages of sand, clay and silt which determined the soil texture. The procedure is given in detail in the Methodology section.

2) Line 31 p. 1- Is the main application this methodology is being created for rainwater harvesting? Reply Information on hydrological soil group is very important in many areas, such as water resources management and planning, agriculture and other engineering projects. However, the main application for this study is the planning of rainwater harvesting. 3) Line 36 p.2- Please provide citation.

Reply (Melesse and Shih 2002). 4) Line 43 p. 2- It is not an alternative to getting a soil type from field and lab data, it is a way to better utilize the data you have. Any remote

sensing model always need actual field and usually lab data to train it.

Reply The authors agree with reviewer that any remote sensing model always need actual field and usually lab data. Therefore, this statement should be replaced by "Remote sensing (RS) represents one of the best solution that may offer possibilities for extending existing soil survey data sets"

5) Line 46 p.2- More appropriate to say "RS can be used to help. Reply Done. 6) Line 55 p.2- Are you actually talking about Remote Sensing here? The paragraph is confusing. Reply The authors completely agreed with the reviewer that the paragraph is confusing therefore, the paragraph should be replaced by "Most reported studies revealed high potential of proximal sensing to estimate soil properties based on clear absorption features at the laboratory and local scale. However, for large scale mapping, this exercise need to be extended beyond the plot scale. Important qualitative and quantitative soil information can be obtained from remote sensing data. Several soil attributes can be determined by spectral analysis under laboratory conditions. However, space borne or airborne spectroscopy complicates the measurements owing to atmospheric influences, lower spectral and spatial resolution, structural effects, spectral mixture of features and geometric distortion (Richter and Schläpfer 2002).

7) Line 58 p.2 – Can be should be used rather than is. Reply Done. 8) Line 58 p.2 – Five specific should be deleted. Reply Done. 9) Line 65 p.2 – What is a large area? Reply The authors will replace 'large area' to 'large scale'. 10) Line 67 p.2 – What correlation? Need to be more specific, and what poor correlation are you referring to? Reply The authors thank the reviewer for this valuable comment. This line is typo error. The authors have decided to delete this line to avoid any confusion regarding the correlation index.

11) Line 90 p.3 – What constitutes a large area? Must be specific about scale? Reply The authors will replace 'large area' to 'large scale'.

12) Line 92 p.3 – Combine the first three paragraphs of this section into one- vey wordy

and broken as written here. Reply Done.

13) Line 94-97 p.3 – delete Reply Done.

14) Line 107 p.4 – Please be more clear here. Are you saying that the shorter valleys are canyons while the longer ones are valleys? How are you defining a canyon? It sounds more like you are trying to describe the variation in the lengths of valleys overall? Reply The authors thank the reviewer for this valuable comment. This is a typo error. The authors have decided to delete this line to avoid any confusion regarding the study area and replace with following paragraph: "The main landscape is a plateau that is divided by Wadi Horan, some of it is canyon- like with a few tens of kilometer long, and others are few hundred kilometers in length drain into Wadi Horan."

15) Line 110 p.4 – Please be more specific, rocky soils? Lots of bedrock outcrops. Reply The major plateau of the study is rocky soil. 16) Line 111 p.4 – What is structure? Reply The dip of the strata is almost horizontal, reaching 1Íẹ to 2Íẹ. The gentle plain reflects the structural position of the study area within the Stable Shelf ( Sadooni 1996).

17) Line 113 p.4 – Do you mean deeper soil or proportion of ground soil? Reply It means proportion of ground soil.

18) Line 117 p.4 – This seems out of place, from the earlier mention of water harvesting, it seems that this is the target application for your method. More explanation is needed to related the two and talk more about water harvesting. Reply The authors completely agreed with the reviewer that more explanation about water harvesting will be added to the revised manuscript "West desert of Iraq is one of the biggest arid regions that has suffered from a severe water shortage, due to its climatic condition and lack of water resources planning and management. When the data are limited or of low quality, decisions related to the planning of rainwater harvesting structures, particularly in the developing countries become more difficult to be made. The nature of most arid regions is generally characterized by the lack of precipitation, high temperature and evaporation, as well as limited surface water and groundwater resources. A rainwater harvesting structure is considered as one of the best solutions to conserve this precious natural resource in the area which has direct effects on both socio-economic development and ecosystem health".

19) Line 119 p.4 – Combine with next paragraph. Reply Done.

20) Line 129 p.4 – Also need a subset map showing the location of the sampling area relative to the larger study area? Reply Please refer to Figure 3

Figure 3 Location of sampling areas

21) Line 130 p.4 – How it is useful? Reply The primitive map provides a good depiction of some spectral classes and categorized these classes based on the ranges of the image value. This depiction is useful to determine and distribute different kinds of soil properties in this study, which reduces time, labor and cost in the initial stage. 22) Line 140 p.4 – What criteria? Methods must be repeatable. Reply With reference to line 131, the criteria considered for sampling locations are the error in pixel vegetation cover, the spectral signatures urban areas, water, slope, roads, soil roughness, location and topography. 23) Line 156 p.5 – How were manipulated? Reply The following paragraphs will be added to give more information on manipulation: " Arc GIS spatial analyst extension is able to convert the themes, depending on vector features to grids (Huisman and Deby 2009). Additionally, grids can be derived and viewed from various spatial analysis operations. These grid cells have been classified in various ways and different colors were chosen for each class, where they represent the progression of values for a specified data attribute. It is achieved after the raster themes are converted into a shape file, which includes the environmental characteristics that represents the hydrological soil group"

24) Line 161 p.5 – There were never specified any were. Reply With reference to line 131, the criteria that was consider for sampling location are the error in pixel vegetation cover, the spectral signatures urban areas, water, slope, roads, soil roughness, location and topography.

25) Line 163 p.5 – better than what, based on what evidence? How would looking at that classification tell me more than different visualization of landsat. I can tell a lot about desert lithology and soils by looking at the landsat images. Reply The word 'better' will be deleted. The authors meant that using unsupervised classification is able to reduce the error with spectral signature. The unsupervised classification is used to classify the physical characteristic and give us wide range of classes than that described through visualization. 26) Line 168 p.5 – To be honest, this whole paragraph and figure 5 are not explained well here, are not document in the methods section, and make very little sense to me. Reply The sensitivity analysis is a common practice to examine the relationship between input and output parameters and to describe the complicity of this relation. Previous studies such as (Chang and Islam 2000; Proctor et al. 2000; Apan et al. 2002) referred to a relationship between each band with special characteristics of soil and used the sensitivity analysis to validate it. Authors suggest to edit this paragraph to give more information about sensitive analysis. The new paragraph is given below: "Sensitivity analysis is a prerequisite to determine the reliability of the model through assessment of uncertainties in the output result (Crosetto and Tarantola 2001). Sensitivity analysis is crucial to test the robustness of input and the extent of output variation when parameters are systematic. For this study, a sensitivity analysis was carried out to validate the relationship between soil type and spectral reflectance, as shown in Fig. 5. Soil type could not be detected by band 2 (wavelength (0.45–0.51) $\mu$m). Band 9 (1.36–1.38 $\mu$m) and band 7 (2.11–2.29 $\mu$m) were the most sensitive to soil type, particularly silt and sand, whereas clayey soil could be detected by band 6 (1.57–1.65 $\mu$m), band 1 (0.43–0.45 $\mu$m) and band 7. Unfortunately, the spectral reflectance for each range of wavelengths represented by the number of bands has a complex relationship with soil type because all these bands participate in detecting the soil texture, but in different weights because of the mineral content of that soil. Because of the variation in spectral reflectance over bands, a highly accurate model for the estimation of soil type is needed. Therefore, it is important to include all effective bands in the ANN model".

27) Line 180 p.5 – No clue what you are describing here? Reply The neural network in this study has multi output where the summation of predicted output equals to the summation of observed data. Therefore, the overall performance was constant. 28) Line 185 p.5 – Not types, these are 3 properties of the soil? Reply The authors thank the reviewer and the word 'types' will be changed to 'three properties of the soil'.

Please also note the supplement to this comment:
http://www.hydrol-earth-syst-sci-discuss.net/hess-2017-13/hess-2017-13-AC2-supplement.pdf

φ₁(·)
center $x_1$

Input
vector
x

$x_1$

$x_2$

$x_{m_0}$

φ₂(·)
center $x_2$

$w_1$

$w_2$

$w_N$

φ_N(·)
center $x_N$

Σ

Output

$Y$

Input Layer    Hidden Layer    Output Layer

**Figure 1 Structure of Radial Basis Function Neural Network**

**Fig. 1.** Structure of Radial Basis Function Neural Network

[Figure]

Figure 2 Topography of the location used to generate digital soil map and DEM of the study area
generated from

**Fig. 2.** Topography of the location used to generate digital soil map and DEM of the study area
generated from Shutter Radar Topographic Mission data

[Figure]

Figure 3 Location of sampling areas

**Fig. 3.** Location of sampling areas